# Validation of the Polish version of the Motivational Postures (Toward Taxes) Questionnaire

**Sabina Kołodziej**  *

Centre for Economic Psychology and Decision Sciences, Kozminski University, Warsaw, Poland

* skolodziej@kozminski.edu.pl

## Abstract

This article presents a Polish adaptation of the Motivational Postures (Towards Taxes) Questionnaire (MPQ). The MPQ is based on the concept of five tax-related motivational postures (Commitment, Capitulation, Resistance, Disengagement and Game Playing) and consists of 29 items. Three studies validating the Polish version of the MPQ are presented. The first study was conducted with a translated version of the original questionnaire and aimed to verify the factorial validity of this version using confirmatory factor analysis (CFA). Since the factor structure revealed on Australian sample was not reproduced, exploratory factor analysis (EFA) was conducted. Study 2 used CFA to confirm the new structure of the modified version of the questionnaire evident from the Study 1 EFA, and also estimated the reliability and internal validity of the modified version. This resulted in a questionnaire consisting of 20 items and five scales (Moral Duty, Capitulation, Active Resistance, Disengagement and Pleasant Games). The third study tested the questionnaire's construct validity. A theoretical interpretation of the scale is provided.

## Introduction

As stated in the well-known Benjamin Franklin quotation, taxes are an inevitable aspect of our lives. However, people exhibit different behaviours, opinions, and attitudes towards this sphere of public life, their understanding of, and willingness to participate in, taxation systems varying. The psychological factors which determine tax-related attitudes and behaviours have been the subject of many economic, legal and purely psychological studies. Most studies have focused on either single aspects or just a few selected aspects of taxation that may lead citizens to positively or negatively evaluate a taxation system. These analyses have described a number of factors that may play an important role in shaping tax-related attitudes and behaviours, both economic and psychological factors playing a part [1–9]. Psychological studies of attitudes towards taxation and the motivation to comply are common [1,5,7,10,11]. Since the process of shaping attitudes is complex, this depending on a person's psychological characteristics and their assessment and interpretation of the reality which surrounds them, an integrative approach is important. The programme of research initiated by Valerie Braithwaite [12–18], who introduced the term motivational postures towards taxes, is a part of this approach.

**Data Availability Statement:** The data underlying the results presented in the study are available from OSF.IO: DOI 10.17605/OSF.IO/6UR3B.

**Funding:** Study 1 and Study 3 did not receive any specific grant from funding agencies in the public, commercial, or not-for-profit sectors. Study 2 was

financed by the Polish National Science Centre's (NCN) Miniatura scheme: Project Number. 2018/02/X/HS6/01430. The funder had no role in study design, data collection and analysis, decision to publish, or preparation of the manuscript.

**Competing interests:** The author has declared that no competing interests exist.

Motivational postures externalize the opinions, preferences, interests, feelings and beliefs of people regarding various forms of respect for state institutions and approval of governmental actions. Motivational postures are associated with certain patterns of individual responses when people interact with state institutions and can relate to one or many areas of state action. A state's actions are based on its legal authority, this not necessarily equating with social assessments of the state's right to govern an individual or a given group of people. Society evaluates power through the prism of its operation and effectiveness. These assessments change over time and are also subject to social influences by virtue of interactions between individuals [12–16]. Therefore, postures defined in this way are social phenomena arising from people's interactions with other people and institutions, being external manifestations of a person's degree of approval and respect for authority, and determining the manner in which a person presents themselves in specific social contexts. In the social space, postures can be used to communicate the quality of a person's relationship with authority and contribute to shaping these relationships in the future [12,16,17].

The main foundation upon which Braithwaite built her typology of motivational postures is the concept of social distance developed by Bogardus [12,19,20]. According to this concept, people differ from each other in terms of the social distances which separate them from institutions of power, other people, and other social groups. The notion of social distance is often used in works explaining phenomena related to resentment between groups, lack of respect, lack of cooperation and prejudice [21–26]. In the context of taxation, it is associated with a person's willingness to cooperate with their tax authority and the person's approval of the authority's activities or, in contrast, a person's unwillingness to participate in a tax system and react negatively to actions taken by their tax authority. Changes in a person's preferred social distance adjust their tax-related activities to best protect their interests. This constitutes a mechanism (increasing one's preferred social distance) by which an individual can situate themselves outside a tax system (or any other system), allowing them to be insubordinate without fearing the possible consequences of their misbehaviour. At the same time, institutions representing the state prefer citizens to be less socially distant as this results in greater citizen responsiveness to actions taken by such institutions [12,14,16]. In this light, motivational postures constitute signals sent by an individual to an authority, other people and themselves, providing information about the distance the person prefers with respect to the authority in question. The existence of these postures has been empirically confirmed in the domain of taxation, and equivalent postures have been described in relation to other areas of state activity such as child protection, occupational health and safety, peace-building activities, nursing home regulation, and agriculture [16,18,27–29].

The second assumption on which Braithwaite based her typology is that state power and legal regulation threaten individual freedom [18], the imposition of regulations by authorities putting at risk the self-esteem of the individual as a being capable of correct, independent and responsible conduct. Braithwaite refers to literature in the field of criminal justice on feelings of shame and humiliation, which shows how the moral self is threatened by the actions of authorities [30–35]. Regulations may also threaten people's feelings of being a member of a community which respects the needs and aspirations of its members. This is especially important when people judge authorities' decisions as unfair because this causes a sense of danger and is interpreted as a loss of individual autonomy [35]. Another area of freedom that authorities may threaten relates to the needs and aspirations of the individual, the realisation of these being restrained due to the obligation to comply with legal norms [18]. The above areas of freedom refer to three areas of the self: the moral self, the self as a member of the democratic collective, and the self as it relates to the pursuit of individual aspirations (the status seeking self). Braithwaite [18] postulates that people cope with the sense of danger caused by authorities'

actions by increasing their social distance, which allows them to regain their lost sense of free-dom. In this light, postures are conceived as motivational, because there is a reason (motiva-tion) for their existence: striving to defend the self against the abuse and extortion perpetrated by authorities. In contrast to the more traditional meaning of the term "motive", the individu-al's response is not hidden, but socially visible, as is its source. At the same time, these postures are shaped by the personal values, beliefs and experiences of the individual, and by social influ-ences [18].

The above assumptions, and their empirical verification in studies of Australian taxpayers, led to the identification of four basic motivational postures towards taxes: commitment, capit-ulation, resistance and disengagement. A fifth motivational posture–game playing–was added by the author after discussions with tax officials about taxpayers' behaviours [12,16–18,36]. The first two postures–commitment and capitulation–refer to beliefs which are consistent with tax authority expectations, defined by Losoncz [37] as adaptation to system's require-ments. When expressed by a person, commitment involves the view that a tax authority's goals are valid, and that a tax system is working properly and should be supported by all citizens. A posture of commitment is associated with a sense of being morally obliged to act for the com-mon good, an example of which is paying taxes conscientiously. Understood in this way, a commitment posture can lead to more active participation in a system, by undertaking activi-ties that go beyond the obligations of the individual. This is motivated by a desire to support the achievement of goals considered to be common. The second posture, capitulation, is also considered to be positive and indicates readiness to act in accordance with an authority's requirements, but without striving to understand, or necessarily sharing, the authority's goals and aspirations. This posture is based on acceptance of the authority, which empowers it to establish the rules of conduct to be followed. With respect to taxation, this means acceptance of a tax administration authority and its ability to enforce the payment of taxes by virtue of its perceived strength. According to Braithwaite [16,18], the commitment and capitulation pos-tures are probably supported by the moral self, and these postures treat compliance with appli-cable regulations as a guarantee of good relations with the institutions of power. Braithwaite [12] expressed the opinion that these two deference postures are the most common in demo-cratic countries.

In contrast to the above positive motivational postures towards taxes, two defiance pos-tures–resistance and disengagement–communicate reluctance to act in accordance with authority expectations. Here, people send signals of unwillingness to comply voluntarily with the demands of institutions of power and other members of a given community. Resistance is associated with the expression of resentment and hostility towards agents of authority, while at the same time accepting that they have the power to enforce cooperation. This posture is also called passive defiance because it does not deny the need for a system. In this case, hostility and resentment result from dissatisfaction with decisions issued by authorities and the way they use power. The resistance posture also includes an expectation that authorities should modify their ways of acting and their systems to respect the social contract between citizens and the state. However, a taxpayer exhibiting this posture is distrustful of the tax system and tax authority, assuming that the latter's main purpose is to detect errors and irregularities in taxpayers' conduct. When dealing with a tax authority, such a taxpayer will strive to minimize their tax burden, and a lack of trust in the tax authority will motivate them to be vigilant when fulfilling their tax obligations, to fight for their rights, and to demand limits to the tax author-ity's power.

The second defiance posture, disengagement, attributes neither authority nor purpose to an extant system. This posture can therefore be defined as an attitude of anomie in which people feel disconnected from the norms and values of a given system and do not pay attention to the

requirements of the system's regulator. According to Durkheim's [38] understanding of anomie, this is not just ignorance of authority expectations, it is often a consequence of a discrepancy between a regulator's rigid norms and an individual's circumstances and accepted social norms. Thus, with respect to taxation, the main goal of a taxpayer exhibiting a posture of disengagement is to maintain social distance from a tax authority and to remain outside the tax system.

The proposition of a fifth motivational attitude–game playing–resulted from Braithwaite's communications with tax administration officials and taxpayers. Therefore, unlike the other four postures, this posture has not been empirically verified in other areas of state activity [12]. On the other hand, similar behaviours have been previously described by researchers dealing with issues related to regulations in the field of economics [39]. McBarnet [40] views game playing as a specific type of approach to legal issues, the law being seen as something that should be tailored to one's own goals and not as something that an individual should respect because it sets the limits of acceptable activity. This type of motivational posture involves a competitive attitude which is aggressive towards tax systems and the authorities enforcing them. Taxpayers adopting this posture focus on using tax laws to their own benefit. They treat their relationship with their tax office as a game, this giving scope for creative interpretation of applicable laws, with little respect being shown for the spirit of these laws [40]. The inclusion of game playing as a fifth motivational posture provides an important means of checking whether people consciously adopt such a style of behaviour in their dealings with tax systems and tax authorities.

Each of the motivational postures that taxpayers can adopt reflects the social distance between an individual and their tax authority and tax system. A smaller social distance will be associated with postures of deference: commitment and capitulation. Contrasting with these are the three postures of defiance: resistance, disengagement and game playing. Resistance and disengagement are closely related to the perception of tax-related issues as being a threat to individual freedoms, low satisfaction with the quality of democracy, anti-government and market-oriented attitudes, relatively weak identification with a given country and one's self-image as an honest taxpayer, an above average desire to use aggressive tax avoidance options, and a desire to abolish the tax system. Postures of defiance are difficult to change because they are characterized by low openness to persuasion in the form of both educational activities and the provision of information, as well as a lack of fear of possible punishment for transgressing laws [12,41].

The above conceptualisation of motivational postures towards taxes assumes that postures can be combined, and that the dominance of any particular posture can change depending on tax authority actions. For example, one might support an authority's pursuit of goals one thinks are right, but one may not agree with the methods chosen to achieve these goals. One might also comply with the regulations of an authority whose actions and aspirations seem indifferent to one's interests or play games with a tax system because this is the norm in one's social group. Thus, the attitudes at issue are not separable, and their intensity may change under the influence of external factors relating to tax authority behaviours and other people and groups of people, as well as internal factors such as our preferences, needs, values, emotions and experiences [16].

Motivational postures towards taxation reflect favourable or hostile feelings of citizens with respect to a state's actions in the sphere of taxation. These attitudes manifest themselves in a willingness to participate in, or desire to be outside, a tax system. Braithwaite's typology has led Australian tax officials to diversify their enforcement strategies to accord with the dominant motivational postures adopted by individual taxpayers. This illustrates the important role that individual differences in tax attitudes can have in shaping interventions aimed at

improving tax compliance [18]. However, it is difficult to find examples of previous experimental studies that incorporate the above diversity, and the adaptation of Braithwaite's questionnaire measuring the described motivational postures will allow researchers to fill this gap with respect to Polish citizens' taxation-related postures.

## Measuring Motivational Postures (Towards Taxes): The original questionnaire

Braithwaite [12] developed a self-report questionnaire to measure individual differences in the five previously discussed motivational postures towards taxation. Most of the questionnaire's items were modelled on questions used in studies of motivational attitudes with respect to other areas of state activity. But items related to game playing were the result of discussions with Australian tax officials about their tax system and the functioning of their tax offices. Thus, as previously mentioned, the genesis of these latter items was specifically related to taxation [12,36]. The questionnaire consists of 29 items, which are affirmative sentences regarding beliefs about a tax system and the way its associated tax offices work. People respond to these statements on a 5-point Likert scale (1 –strongly disagree, 5 –strongly agree). The original version of the questionnaire contains the following numbers of items for each scale: Commitment, 8 items; Capitulation, 5 items; Resistance, 6 items; Disengagement, 5 items; Game playing, 5 items.

Ratings for a specific motivational posture are summed and then divided by the number of items in the scale, therefore scores for each posture range from 1 to 5. PCA results have shown that each of the factors is defined predominantly by statements representing one of the postures [36]. This suggests that the motivational postures tapped are relatively distinct. Levels of internal consistency are also satisfactory, Cronbach's α coefficients for the original version of the questionnaire ranging from .63 to .82 (Commitment: .82; Capitulation: .63; Resistance: .68; Disengagement: .64; Game Playing: .69). These observations provide evidence that the different postures constitute reasonably coherent sets of beliefs which form part of the way that people think about themselves in relation to tax authorities [12,36]. While the questionnaire was developed on, and initially used with, Australian taxpayers, subsequently it has been used in studies in other English-speaking countries [9,42,43].

## The Motivational Postures (Towards Taxes) Questionnaire (MPQ)– a Polish adaptation

Presently, three studies were conducted to validate a Polish version of the MPQ. A first study aimed to verify the factorial validity of a translated version of the original questionnaire using confirmatory factor analysis (CFA). Since the factor structure described by Braithwaite [12,18] was not reproduced in the CFA, explanatory factor analysis (EFA) was conducted on the same data. In a second study, further data were collected using a reduced version of the translated questionnaire and these data were again subjected to CFA to confirm the questionnaire's structure as established by the EFA in the first study. The second study also assessed the reliability and internal validity of the modified version of the questionnaire. A third study tested the modified questionnaire's construct validity.

### Study 1

Study 1 aimed to examine the factor structure of the translated version of the MPQ. Initially, CFA with maximum likelihood (ML) estimation using AMOS 21 was performed. Because this analysis did not confirm the original questionnaire's factor structure for the Polish version, in a second analysis, EFA was conducted to establish the factor structure for this version.

**Participants.** There were 600 participants in the first study (314 women, $M_{age}$ = 31.39 years, $SD_{age}$ = 10.56 years). The majority of participants were professionally active (78.3%) at the time of the study, either as employees (69.7%) or entrepreneurs (8.7%). Almost all participants (96%) were active taxpayers submitting an annual tax return in the year preceding the survey. The study was carried out using an online university research panel via LimeSurvey. Participation in the study was voluntary and no renumeration was paid. Participants written consent was collected in the research panel.

**Methods.** After obtaining the original author's approval for the questionnaire's adaptation, a Polish-language version of the questionnaire was developed. This Polish version was created by first translating the English version into Polish and then back translating this version into English. Translation accuracy was assessed by four competent judges (two psychologists and two economists). Subsequent to this preparation, participants completed the questionnaire to provide the data for Study 1. The Human Research Ethics Committee of the Kozminski University approved the study, which was carried out in accordance with the Board's recommendations.

**Results.** *The factor structure of the translated questionnaire*. CFA showed that a five-factor model provided a poor fit to the data. The value of the RMSEA (*Root Mean Square Error of Approximation*) indicator, which is an index of the approximate error variance in a model, is the most frequently used fit indicator. Values closer to 0 represent a better fit and should be below .08 [44,45], but for the model tested RMSEA had a value of .088 (PCLOSE = .001) and was therefore not within the bounds of acceptability. Other fit indices were also unsatisfactory. For example, AGFI (*Adjusted Goodness of Fit Index*) and CFI (*Comparative Fit Index*), which in the case of a good fit should have values no lower than .90 [44,45], were .735 and .647 respectively and were therefore unacceptable. So, the CFA showed that the original factor structure established using Australian taxpayers data was not replicable for the translated instrument.

Given the above conclusion, the translated questionnaire's factor structure was determined by performing EFA (specifically, Principal Axis Factoring using Varimax rotation and Kaiser normalisation) on the Study 1 data. The obtained solution showed a six-factor structure, with only one item, which was not correlated with other items in the questionnaire, loading on the sixth factor. Moreover, low representativeness (factor loadings below 0.35) was observed in the case of three other items. Therefore, these four items were excluded from a second EFA, the solution for which is presented in Table 1.

Table 1 shows the items of the questionnaire in their original form [12]. Loadings of items that were included in the final version of the questionnaire are indicated in bold type.

The results of the final EFA showed that 25 items loaded on five factors, and the majority of factor loadings for items in the Polish version of the questionnaire were consistent with Braithwaite's [12] proposed factor structure. Seven items loaded on the first factor, which accounted for 21.75% of item variance. Six of these items were from the Commitment scale of the original version of the questionnaire. Five items loaded highly on the second factor and accounted 12.06% of item variance. Four of these five items were from Braithwaite's Disengagement scale. Five motivational posture items loaded on the third factor, which accounted for 7.85% of item variance. Four of these items were from the Capitulation scale of the Australian version of the MPQ. Five items loaded highly on the fourth factor, which accounted for 5.47% of item variance. Four of these items were included in the Resistance scale of the original MPQ. Finally, only three items loaded on the fifth factor, which accounted for 4.71% of item variance. These three items were from the original instrument's Game playing scale. Summarizing, more than 51% of item variance was accounted for by the five-factor EFA solution. The lowest factor loading was .39 and the average factor loading was 0.62. Therefore, it can be

**Table 1. Item loadings for the exploratory factor analysis of the Polish version of the questionnaire.**

| | Factor 1 | Factor 2 | Factor 3 | Factor 4 | Factor 5 |
|---|---|---|---|---|---|
| *Commitment* | | | | | |
| Paying tax is the right thing to do. | **.78** | | | | |
| Paying tax is a responsibility that should be willingly accepted by all taxpayers. | **.81** | | | | |
| I feel a moral obligation to pay my tax. | **.75** | | | | |
| Paying my tax ultimately advantages everyone | **.60** | | | | |
| I think of tax paying as helping the government do worthwhile things. | | | .51 | | |
| Overall, I pay my tax with good will. | .39 | | | | |
| I accept responsibility for paying my fair share of tax. | **.68** | | | | |
| *Capitulation* | | | | | |
| If you cooperate with the Tax Office, they are likely to be cooperative with you. | | | .46 | | |
| Even if the Tax Office finds that I am doing something wrong, they will respect me in the long run as long as I admit my mistakes. | | | .48 | | |
| The Tax Office is encouraging to those who have difficulty meeting their obligations through no fault of their own. | | | .74 | | |
| The tax system may not be perfect, but it works well enough for most of us. | | | .61 | | |
| No matter how cooperative or uncooperative the Tax Office is, the best policy is to always be cooperative with them. | .48 | | | | |
| *Resistance* | | | | | |
| The Tax Office is more interested in catching you for doing the wrong thing, than helping you do the right thing. | | | | .56 | |
| It's important not to let the Tax Office push you around. | | | | **.69** | |
| It's impossible to satisfy the Tax Office completely. | | **.58** | | | |
| Once the Tax Office has you branded as a non-compliant taxpayer, they will never change their mind. | | | | .52 | |
| As a society, we need more people willing to take a stand against the Tax Office. | | | | **.59** | |
| *Disengagement* | | | | | |
| If I find out that I am not doing what the Tax Office wants, I'm not going to lose any sleep over it. | | **.66** | | | |
| I personally don't think that there is much the Tax Office can do to me to make me pay tax if I don't want to. | | .57 | | | |
| I don't care if I am not doing the right thing by the Tax Office. | | **.54** | | | |
| If the Tax Office gets tough with me, I will become uncooperative with them. | | **.66** | | | |
| I don't really know what the Tax Office expects of me and I'm not about to ask. | | | | .56 | |
| *Game Playing* | | | | | |
| I enjoy talking to friends about loopholes in the tax system. | | | | | **.79** |
| I like the game of finding the grey area of tax law. | | | | | **.76** |
| I enjoy the challenge of minimising the tax I have to pay. | | | | | .57 |

concluded that the five-factor structure of the original Australian version of the questionnaire was reproduced, but with slightly different items assigned to each of the scales. This reduced version of the questionnaire was used to conduct Study 2.

## Study 2

The second study was conducted to validate the new version of the questionnaire resulting from the Study 1 analyses. Moreover, Study 2 allowed descriptive statistics for the newly validated Polish version of the MPQ to be produced along with statistics establishing the instrument's reliability and internal validity.

**Participants.** Study 2 was conducted on a representative sample of 1097 Polish taxpayers (523 men, $M_{age}$ = 45.23 years, $SD$ = 15.81 years). More than half of the participants (56.9%) were professionally active, either as employees (47.6%) or entrepreneurs (9.3%). Almost a quarter of the respondents (24.5%) were retired pensioners, which is a consequence of the age structure of Polish society. Only 5.7% of respondents were students. The vast majority of

respondents (91.2%) were active taxpayers submitting an annual tax return. Participation in the study was voluntary and participants received no renumeration. Participants written consent was collected in the research panel. The study was realized using the Ariadna Polish national research panel, respondents completing the survey anonymously. After analysing survey completion times to identify and eliminate outliers (defined as respondents taking less than four minutes or more than 20 minutes to complete the questionnaire), data for 942 respondents were included in the analyses reported.

**Methods.** Based on the Study 1 results, data for a 25-item questionnaire were collected with a view to performing CFA to verify a new five-factor model of motivational postures towards taxation. In this model, the four items previously shown to correlate very weakly with other MPQ items were removed. The Human Research Ethics Committee of the Kozminski University approved the study, which was carried out in accordance with the Board's recommendations.

**Results.** *A new five-factor model of motivational postures towards taxation.* The Study 2 CFA was again performed with AMOS 21 using maximum likelihood estimation. For the initial five-factor model tested, the RMSEA value was .064 (PCLOSE = .001) and the AGFI and CFI coefficients were .87 and .90 respectively. Thus, while these indices indicated a better fit than those for the Australian five-factor model in Study 1, they did not permit the conclusion that the model tested was satisfactory. To improve the fit of the model, its modification indices and standardized residual covariances were examined. This examination led to the removal of five items from the model and to covariances being added between items within one factor. Then CFA was performed again. The results for this model indicated a satisfactory fit between the model and the data: RMSEA = .060 (PCLOSE = .001); AGFI = .91, CFI = .93.

The above process resulted in a questionnaire consisting of 20 items and five scales. Although the number of scales in the Polish version of the MPQ is consonant with the original version, the composition of items forming the individual scales is different. Not only have a few items been assigned to different scales, but nine items have also been removed in the final version of the Polish MPQ adaptation. The existence of these differences has necessitated the proposal of new names for three of the five scales in the modified instrument, these names being based on factor loadings for items. Thus, Moral Duty (rather than Commitment, as in Braithwaite's original 2002 instrument) seems to be a reasonable title for the 5-item scale describing the most positive motivational posture toward taxes, as the majority of items in this scale refer to the moral obligation to pay taxes. Also, the Resistance scale has been renamed Active Resistance as, of the four items constituting the scale, the two items with the highest loadings place an emphasis on activities performed when in contact with tax administrators. Finally, it is apposite to name the instrument's fifth scale Pleasant Games, this emphasising the positive emotions a taxpayer is likely to experience when playing games with their tax office to achieve a favourable result. Only two of the five items in Braithwaite's [12] original Game Playing scale are present in this scale. The names of the two remaining scales: Capitulation (five items) and Disengagement (four items) have been left unchanged as they adequately describe the scales' theoretical content.

*Reliability analysis.* The next step in validating the MPQ was to perform reliability analysis of the final version of the questionnaire. The following Cronbach's α reliability coefficients for the five motivational postures towards taxes scales were obtained: Moral Duty, α = .87; Capitulation, α = .74; Active Resistance, α = .71; Disengagement, α = .76; Pleasant Games, α = .65. The first four of these coefficients indicate good reliability for the scales involved and are higher than those for the equivalent scales in Braithwaite's [12] original questionnaire. The coefficient for the Pleasant Games scale, however, is slightly lower than desired, this being likely to be because this scale has only two items, which makes it difficult to obtain an

**Table 2. Pearson's r correlations showing relationships between scores on the five scales.**

|  | Moral Duty | Capitulation | Active Resistance | Disengagement | Pleasant Games |
|---|---|---|---|---|---|
| Moral Duty | - | .70** | -.35** | -.30** | -.04 |
| Capitulation | | - | -.25** | -.15** | .04 |
| Active Resistance | | | - | .60** | .37** |
| Disengagement | | | | - | .46** |
| Pleasant Games | | | | | - |

**p < .01.

acceptable alpha coefficient. Future research on the questionnaire might add items to this scale, or, alternatively, perhaps the scale should simply be removed from the instrument.

*Internal validity analysis.* The questionnaire's internal validity was assessed by computing Pearson's *r* correlations between participants' scores on its scales (see Table 2). Here, the highest coefficient was obtained for the two scales measuring positive motivational postures towards taxes (Moral Duty and Capitulation), this being followed by the two scales indicating negative motivational postures (Active Resistance and Disengagement). As would be expected, significant negative correlations were observed between these two types of scale (Moral Duty and Capitulation vs. Active Resistance and Disengagement). The latter two scales were significantly positively correlated with the Pleasant Games scale, but this latter scale was not significantly correlated with either the Moral Duty or the Capitulation scale. In their entirety, then, these results vouched for the questionnaire's internal validity.

*Descriptive statistics.* To conclude the psychometric analysis of the questionnaire performed in Study 2, descriptive statistics for the instrument's four scales are presented in Table 3. Since the number of items in scales varies, statistics are based on average scores for items forming scales, rather than summated responses to items.

Table 3 shows that the Moral Duty and Capitulation scales produced the highest scores for the present Polish sample, this being in line with Braithwaite's [12,18] findings that deference postures are the most commonly observed postures. The Active Resistance scale exhibited a slightly lower mean and the means for the Pleasant Games and Disengagement scales were the lowest.

## Study 3

In Study 3 a further investigation of the Polish version of the Motivational Postures (Towards Taxes) Questionnaire was performed. To evaluate the construct validity of this Polish version, four psychological constructs were chosen: contingencies of self-worth, individual and social tax standards, propensity to take financial risks, and belief in life as a zero-sum game.

According to Crocker and Wolfe's [46] contingencies of self-worth concept, self-worth should be analysed in relation to the beliefs a person has about themselves in life domains

**Table 3. Descriptive statistics for the questionnaire's scales.**

|  | *M* | *SD* | Skewness | Kurtosis |
|---|---|---|---|---|
| Moral Duty | 3.53 | .76 | -.61 | .54 |
| Capitulation | 3.25 | .69 | .40 | .82 |
| Active Resistance | 3.17 | .67 | .31 | .59 |
| Disengagement | 2.75 | .73 | .33 | .56 |
| Pleasant Games | 2.79 | .87 | .17 | .06 |

which can be a source of self-esteem. Using Crocker et al. [47] Contingencies of Self-Worth Scale (CSWS), two contingencies of self-worth were considered in the present study: virtue and approval from others. The first construct–virtue–reflects internal, autonomous aspects of the self, while the second–approval from others–concerns self-worth based on social standards and arises from social comparison processes. Virtue concerns one's moral adequacy and leads one to define oneself as a good, moral and worthwhile person [47]. Therefore, people's scores on the CSWS Virtue scale should be positively related to their scores on the MPQ Moral Duty scale where a high score indicates that a person believes that tax compliance is one of the elements of overall moral adequacy. Obedience to rules is also tapped by the Capitulation scale, where the emphasis is on cooperation with the tax authorities, therefore this scale should exhibit a positive correlation with the CSWS Virtue scale. Moreover, no correlation between Virtue scores and MPQ Pleasant Games scale scores should be observed since the latter scale measures a person's tendency to follow taxation rules without respecting the spirit of the law.

The second CSWS scale currently utilised–Approval from Others–assumes that other people's views are an important basis of one's self-esteem. Therefore, compliance with tax regulations may contribute to shaping positive self-esteem, while for people adopting a negative motivational posture towards paying taxes this contingency should be unimportant. Given these assumptions, scores on the Approval from Others CSWS scale should positively correlate with MPQ Moral Duty and Capitulation scores but negatively correlate with scores on the remaining MPQ scales.

Moving on, Niesiobędzka [48] considered tax evasion tendencies in the context of individual and social standards that may strengthen or weaken such propensities. These standards describe personal rules and perceived social norms regarding different types of tax evasion and can be measured using Niesiobędzka's [48] 11-item questionnaire, which consists of two scales: (1) Personal standards regarding different ways of evading tax, and (2) Social norms regarding tax evasion. According to the author, if one's social observations suggest that it is socially acceptable to act unlawfully with respect to paying taxes, this may reduce one's propensity to comply. It is reasonable to assume that these standards correlate with motivational postures towards taxation, greater personal and social consent with respect to breaking tax rules being likely to be associated with negative motivational postures towards taxation. Conversely, as individual and social standards condemning tax evasion contribute to higher tax compliance [48] they may also strengthen positive motivational postures towards taxation. Therefore, as Niesiobędzka's questionnaire measures individual and social norms conducive to tax evasion, negative correlations with the MPQ's Moral Duty and Capitulation scales were hypothesized, and positive relationships were hypothesized with the two MPQ scales measuring negative motivational postures towards taxes: the Active Resistance and Disengagement scales. No significant correlation between individual and social standard scores on Niesiobędzka's instrument and MPQ Pleasant Games scale scores was expected since a person's tendency to follow taxation rules as measured by the latter scale should not be linked to internal or external standards but, rather, such a tendency should be linked to a desire to maximize one's own benefits.

The propensity to take financial risks, as measured by the Financial Risk subscale of Studenski's [49] Multi-Factor Risk Behavior Scale (WSZR), was another construct used to establish the MPQ's validity. Here, it was thought reasonable to assume that there should be a positive relationship between scores on the MPQ Pleasant Games scale and financial risk-taking: searching for legal loopholes and a competitive aggressive attitude towards a tax system and tax administration should be associated with a propensity to take risky financial decisions.

Finally, belief in life as a zero-sum game is the conviction that one person's gains are made at the expense of other people [50]. People expressing this belief claim that a person's

economic success is only possible if someone else experiences a loss, implying that people's interests are inherently antagonistic. Other people are therefore seen as selfishly oriented and this acts as a deterrent to cooperation. In the current validation study it was assumed that belief in life as a zero-sum game (as measured by Różycka and Wojciszke's [50] eponymous scale) should be associated with negative postures towards taxation (Active Resistance and Disengagement) and a person's desire to minimize their tax burden: people with negative motivational postures towards taxation are unwilling to support state finances since they resent, and are hostile towards, authority. Similarly, belief in life as a zero-sum game should be positively correlated with Pleasant Games scale as this posture reflects person's selfish attitude towards the tax system. Moreover, it was expected that scores on the Belief in Life as a Zero-Sum Game scale should be negatively correlated with those on the Moral Duty scale. This MPQ scale measures the conviction that paying taxes is an obligation common to all citizens and leads to a win-win situation: a view which is contrary to that measured by Różycka and Wojciszke's [50] scale.

**Participants.** There were 313 participants in Study 3 (152 women, $M_{age}$ = 21.39, $SD_{age}$ = 2.41). Since this study was conducted with students from two different universities as participants, no data regarding professional activity and submitting an annual tax return were collected. Participation was voluntary and no renumeration was given. Participants written consent was collected in the research panel. As in Study 1, the study was conducted using an online university research panel via LimeSurvey.

**Methods.** As previously mentioned, four psychological constructs were adopted as criteria to assess the modified MPQ's construct validity: Propensity to take financial risks, contingencies of self-worth, individual and social tax standards, and belief in life as a zero-sum game. These constructs were measured using scales with proven psychometric characteristics:

- The Financial Risk subscale of the Multi-Factor Risk Behavior Scale (WSZR) [49]

- The Virtue and Approval from Others scales of a Polish version of the Contingencies of Self-Worth Scale (CSWS) [51]

- The Individual and Social Tax Standards Scale [48]

- The Belief in Life as a Zero-Sum Game Scale [50]

Given the total length of the questionnaires used and the possible impact of respondent fatigue on the data obtained, respondents were randomly assigned to one of two groups: Group 1 completed the MPQ, Multi-Factor Risk Behavior Scale [49] and Contingencies of Self-Worth Scale [51], while Group 2 completed the MPQ, Individual and Social Tax Standards Scale [48] and Belief In Life As A Zero-Sum Game Scale [50]. The Human Research Ethics Committee of the Kozminski University approved the study, which was carried out in accordance with the Board's recommendations.

**Results.** *Construct validity analysis.* To assess the construct validity of the Polish version of the MPQ, Pearson's *r* correlation analysis was performed to test for relationships between scores on the MPQ's four scales and scores on the above-mentioned psychological instruments (see Table 4).

Table 4 shows that, as expected, the Virtue scale of the Contingencies of Self-Worth Scale exhibited significant positive correlations with the MPQ Moral Duty and Capitulation scales but no significant relationship with the MPQ Pleasant Games scale. Moreover, as also expected, there were negative correlations between the CSWS Virtue scale and the MPQ scales expressing negative motivational postures towards taxes (Active Resistance and Disengagement). The second CSWS scale–Approval from Others–correlated significantly with all the

**Table 4. Pearson's r coefficients for relationships between scores on the five Polish MPQ scales and scores on the four validation instruments.**

| | Contingencies of Self-Worth Scale | | Individual and Social Tax Standards Scale | | Belief in Life as a Zero-Sum Game Scale | Multi-Factor Risk Behavior Scale |
|---|---|---|---|---|---|---|
| | Virtue | Approval from Others | Individual standards | Social norms | | Financial Risk |
| Moral Duty | .39** | .31** | -.49** | -.23** | -.21** | -.07 |
| Capitulation | .25** | .27** | -.50** | -.32** | -.07 | -.07 |
| Active Resistance | -.35** | -.33** | .67** | .40** | .23** | .06 |
| Disengagement | -.40** | -.35** | .64** | .34** | .15* | .02 |
| Pleasant Games | -.05 | -.24** | .11 | .14 | .16* | .35** |

*$p < .05$

**$p < .01$.

MPQ scales, the directions of the relationships being as proposed: positive correlations with the MPQ scales measuring positive motivational postures towards taxes (Moral Duty and Capitulation) and negative correlations with the remaining scales.

Correlations for the Individual and Social Tax Standards Scale were also consistent with the theoretical assumptions made. Both, individual standards and social norms accepting tax cheating correlated negatively with the MPQ Moral Duty and Capitulation scales but positively with the MPQ Active Resistance and Disengagement scales. As expected, neither individual standards nor social norms was significantly correlated with the MPQ Pleasant Games scale.

The Belief in Life as a Zero-Sum Game scale was positively correlated with three MPQ scales: Active Resistance, Disengagement and Pleasant Games. Again, the relationships were expected to occur. Further, the MPQ Moral Duty scale correlated negatively with the Belief in Life as a Zero-Sum Game scale, there was no significant correlation between the latter measure and the MPQ Capitulation scale, both of these observations being as anticipated.

The Financial Risk subscale of the Multi-Factor Risk Behavior Scale only correlated significantly with the MPQ Pleasant Games scale. In addition to the lack of significant correlations involving the other MPQ scales, the negative character of this relationship confirmed expectations regarding the Financial Risk measure.

## Discussion

This article has recounted the adaptation for Polish use of Braithwaite's questionnaire for measuring motivational postures towards taxes [12–16]. It was found that factors describing the instrument's original structure were not reproducible in a Polish sample of respondents.

Statistical analysis led to reproduction of the five-factor structure of the questionnaire, albeit that it was necessary to make changes in the content of the original questionnaire's scales by removing nine items and reassigning some items to different scales. Thus, the Polish version of the questionnaire measures five motivational postures towards taxation, the theoretical interpretations of the individual scales being somewhat modified.

The Moral Duty scale taps the belief that paying taxes is a moral and civic duty, reflecting a person's sense of social responsibility: the view that the existence of a tax system benefits all citizens, and that all members of society should agree to pay taxes where they are due. As in the original version of the MPQ, this scale reflects the most positive attitude towards taxes and tax administration, conceiving willingness to pay taxes as deriving from a desire to support the achievement of society's common goals. However, the fact that the Commitment scale in the original Australian questionnaire consisted of items that were somewhat different from those

loading on the roughly equivalent Polish factor resulted in a change of name for the Polish scale. The naming of this Polish scale reflects an emphasis on a person's sense of duty but not a propensity to undertake activities which go beyond a person's obligations as stated in Braithwaite's works [12,16–18,36].

Another scale indicating positive motivational posture toward taxes and tax system is Capitulation. People scoring high on this scale believe that, while the tax system they have to deal with may not be perfect, everyone should accept that they need to pay taxes for the greater benefit of all members of society. High scores on the scale also indicate a positive attitude towards tax administrators, cooperation with one's tax office being seen as personally beneficial, particularly in the event that an error is made.

High scores on the Active Resistance scale reflect a strongly antagonistic attitude towards the tax system and its administration. Such an attitude involves a "cops and robbers" philosophy concerning the citizen–tax authority relationship, based on the assumption that tax authorities are likely to treat taxpayers as potentially fraudulent. Thus, a person with this attitude has the perception that a tax office is likely to be more interested in catching a taxpayer for any transgressions rather than helping them to be compliant. Moreover, these mutually antagonistic positions are perceived as stable. Finally, a state of active resistance is also characterized by the belief that people should oppose tax office decisions whenever possible.

As originally conceived, Disengagement is a motivational posture indicating an individual's desire to maintain a social distance from a tax authority and to remain outside the tax system, and the Polish factor loadings retained the original scale's nature. Thus, an individual exhibiting a posture of disengagement wishes to be outside the tax system and is unresponsive to the requirements of its regulators. Moreover, a person characterized by this motivational posture perceives tax authorities to be weak and unable to force a taxpayer to behave in the ways such authorities desire.

People with a high Pleasant Games score adopt the attitude that it is acceptable for a taxpayer to analyse tax regulations with a view to finding grey areas which enable them to reduce their tax burden [12]. These people also converse with others about ambiguities in tax regulations. Given the positive emotions arising from the playing of such games with the tax system, in the present work these games were labelled as "pleasant".

It should be noted that, to date, no analogous adaptation of Braithwaite's questionnaire has been carried out for any other national group, and that all previous studies have been based on the original English version of the questionnaire. The results of the current series of studies show that Braithwaite's concept of motivational attitudes towards taxation, which was based on research with Australian taxpayers [12–16], needs slight adaptations if it is to be used in other countries. The observed differences in motivational postures among Australian and Polish taxpayers show that these constructs are shaped by both context and person characteristics. In the tax domain the country's tax system and administration is a key contextual factor. However, the reconstructed structure of the MPQ retains the number of factors as well as clear dichotomy of motivational postures towards taxes as indicated by Braithwaite's above referenced studies, as well as their coexistence. As was the case with Australian taxpayers, the current survey on a representative sample of Polish taxpayers showed that an attitude demonstrating a positive posture towards taxation prevails, negative postures occurring to a lesser extent.

Interestingly, in the present study the motivational posture scale concerning taxpayers treating relationships with their tax office as a game focusing on minimizing the amount of taxes paid was reconstructed (although in a reduced version). As noted earlier, unlike the other scales which were derived from statistical analysis of initial research data, this scale's development was based on conversations with Australian tax officials. However, the existence

of this factor was then proved in later studies [36]. The present results show that some Polish taxpayers also treat their tax system in a similar way, deriving satisfaction from the reduced taxes they pay.

The construct validation analyses conducted in Study 3 confirmed the existence of expected relationships between motivational postures towards taxes and others psychological characteristics. It was shown that both MPQ scales measuring positive postures (Moral Duty and Capitulation) correlated positively with both Contingencies of Self-Worth scales (Virtue and Approval from Others), and negatively with the Individual and Social Tax Standards Scale. Additionally, the MPQ Moral Duty scale exhibited a negative relationship with the Belief in Life as a Zero-Sum Game Scale. In addition to confirming the construct validity of the Polish version of the MPQ, these results showed that the concept of paying taxes as an expression of fulfilling one's civic duty, obeying rules and cooperating with authorities are likely to be important in shaping one's self-worth, coming from internal, autonomous aspects of the self and social comparison processes. The previously mentioned negative relationships with individual and social standards concerning tax evasion reinforced these conclusions. The perception that paying taxes is a moral duty was also associated with a propensity to cooperate with others rooted in a conviction that such behaviour leads to outcomes that are beneficial for society as a whole.

Relationships involving the MPQ scales measuring negative motivational postures towards taxes (Active Resistance and Disengagement) where the opposite of those described above. This confirms the dichotomous nature of the postures described in the MPQ. Here, there were significant positive correlations with individual and social standards regarding different types of tax evasion as well as belief in life as a zero-sum game. Thus, as expected, negative motivational postures towards the tax system and it's administration, manifesting themselves in a reluctance to act in accordance with tax authority expectations, were positively linked to individual standards and the perception of social standards suggesting that evading taxes is an acceptable thing to do. Also, declaring an unwillingness to comply voluntarily with the demands of institutions of power and other members of a given community was connected with the beliefs that one person's gains are made at the expense of other people and that people's interests are inherently antagonistic, as measured by the Belief in Life as a Zero-Sum Game Scale. Finally, the negative correlations between the two negative motivational postures and the two CSWS indices showed that not abiding by tax laws and not willingly cooperating with tax authorities were opposed to values shaping people's high self-worth.

As expected, the fifth MPQ scale–Pleasant Games–was significantly positively correlated with the Financial Risk scale, indicating that perceiving relationships with the tax authorities as a game aimed at reducing tax payments via exploitation of ambiguities in tax regulations was associated with the propensity to take financial risks. Pleasant Games scale was also significantly positively correlated with the Belief in Life as a Zero-Sum Game Scale. Both scales involve a competitive attitude and a focus on one's own benefits. Additionally, the fifth motivational posture was negatively correlated with the CSWS Approval from Others scale. This result, although not initially hypothesized, is consistent with the construct which the Pleasant Games scale seeks to measure. According to Braithwaite [12], people manifesting a Pleasant Games posture may perceive others as rivals in the pursuit of limited resources. Not only might such people think it unnecessary for people to approve their actions, they might also view actions which win the approval of others in a negative light, because such actions might reduce their chances of prospering.

The above results of analyses examining relationships between the MPQ scales and variables relating to theoretically similar constructs, and variables that are related to motivational

postures towards taxes, were consistent with expectations. For the most part, the strengths of the hypothesized relationships were at least moderate, and in a few cases quite large.

To conclude, together with CFA indices ultimately indicating a good fit between the model and the Study 2 data, the results of the above construct validation analyses, and the reliability and internal validity analyses also conducted in Study 2, demonstrate that the Polish adaptation of the MPQ has good psychometric qualities on the whole. The only problematic aspect of the questionnaire appears to be the low number of items in the Pleasant Games scale and, concomitantly, this scale's lower than desirable reliability. Future work should rectify this problem, but in the meantime, for the most part, the present version of the MPQ appears suitable for use in research focusing on Polish taxpayers.

## Supporting information

**S1 File. Polish version of the Motivational Postures (Toward Taxes) Questionnaire.**
(PDF)

## Author Contributions

**Conceptualization:** Sabina Kołodziej.

**Formal analysis:** Sabina Kołodziej.

**Funding acquisition:** Sabina Kołodziej.

**Methodology:** Sabina Kołodziej.

**Validation:** Sabina Kołodziej.

**Writing – original draft:** Sabina Kołodziej.

**Writing – review & editing:** Sabina Kołodziej.

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
