## [Decision Letter · Decision Letter 0]

5 May 2021

PONE-D-21-07749

Validation of the Polish version of the Motivational Postures (Towards Taxes) Questionnaire

PLOS ONE

Dear Dr. Kołodziej,

Thank you for submitting your manuscript to PLOS ONE. After careful consideration, we feel that it has merit but it needs proofreading. Therefore, we invite you to submit a revised version of the manuscript. Feel free to reflect on comments from reviewers but mandatory is only fixing typos.

We look forward to receiving your revised manuscript.

Kind regards,

Frantisek Sudzina

Academic Editor

PLOS ONE

Journal Requirements:

3. Your abstract cannot contain citations. Please only include citations in the body text of the manuscript, and ensure that they remain in ascending numerical order on first mention.

Reviewers' comments:

Reviewer's Responses to Questions

**Comments to the Author**

1. Is the manuscript technically sound, and do the data support the conclusions?

Reviewer #1: Yes

Reviewer #2: Yes

2. Has the statistical analysis been performed appropriately and rigorously? 

Reviewer #1: Yes

Reviewer #2: Yes

3. Have the authors made all data underlying the findings in their manuscript fully available?

Reviewer #1: Yes

Reviewer #2: Yes

4. Is the manuscript presented in an intelligible fashion and written in standard English?

Reviewer #1: Yes

Reviewer #2: Yes

5. Review Comments to the Author

Reviewer #1: Review: PONE-D-21-07749 Validation of the Polish version of the Motivational Postures (Towards Taxes) Questionnaire

This paper is extremely clearly written and the research is rigorously undertaken. I have no hesitation in recommending publication. The following comments would, I believe, strengthen the paper. But the quality of the paper merits publication as it stands.

The paper does an excellent job of explaining that both context and person characteristics shape motivational posturing in the introduction. This applies to the strength of the postures and also to the specific forms that the postures take. This means that there are two levels of analysis for a paper such as this. Does the general form of the posturing hold across different tax/regulatory regimes? And if the general form does hold up, what is our process for developing specific items that reflect the particular manifestations of these general postures in particular countries. I would argue they will be different because the context both administrative and legal has its own effect on the specific expressions of the general posture. This is how I understand the differences in commitment/moral duty and gameplaying/pleasant games). This is why, in my view, this paper is so very important. But I think this point is not really made in the paper as it is presented at the moment. Let me explain what I mean below with this example.

In Poland, duty to pay tax is the lynchpin of a commitment to taxation, not for example, willingness to pay more than one has to or do the right thing in paying tax. How do we make sense of this? In Australia, the tax system for employees is pay-as-you-go so when we do our annual tax return, many look forward to receiving a tax refund. Some would say in this situation, “I can’t be bothered finding all my deductions, this will do.” Others would say “I am very careful to pay only the tax I need to”. In other words, paying more than you probably needed to and paying exactly what you need to come together as commitment. If there is no pay-as-you-go system or if you are a small business owner and you are doing your tax return to calculate annually how much you have to pay, there is no little nudge that will lead you to pay more than you need to! It is a big job and that alone will mean you are likely to pay just what you are legally obligated to pay. So moral obligation separates at the level of specific items from I am happy to pay my tax or words to that effect.

Similarly, I would argue that game playing/pleasant games is very much expressed in context – depending on the country’s tax system and administration. The items will be different (and I have experienced similar differences when measuring postures in work health and safety rather than taxation). The basic ideas are the same however – the higher order construct if you like. The pattern of correlations in Study 3 makes me think this is the case.

The authors should only make these changes if they wish – I am not suggesting this is necessary for publication. I think it is just an important contribution for someone to make in this field of research.

There are some specific typos that should be corrected and a few minor points along the way:

Line 300 each instead of ech

Line 316 Should this be anonymously ?

Line 404 al. not all.

Line 509-510 By saying “correlated positively” the author means that having individual and social standards against tax cheating is positively correlated with moral duty etc. The individual and social standards scale are scored so that a high score means tolerance of tax cheating as I read the scale descriptions and the negative correlations in Table 1. It would be desirable to clarify the fact that the way the scale is scored produces the negative correlations, but these negative correlations reflect an association between adopting standards against tax cheating and moral duty.

Line 515 Braithwaite’s work on gameplaying in Defiance in Taxation and Governance clearly shows game playing as a Zero-sum game life view. See SEM analysis in the book on http://valeriebraithwaite.com

Line 556-557 I particularly like the way these data confirm Braithwaite’s values work (again the Defiance book on the website) that there is some stability in these postures. Someone ideologically opposed to tax is not likely to turn into a moral duty advocate for tax because of the values base (represented in this work as philosophy of life).

P606 dichotomous nature of postures. I see this argument. Psychological research on context tells us people are complex however. Post-Covid may be more people will hate the tax authorities but feel a duty to pay. A question for future research!

I enjoyed this paper very much. It was a pleasure to review it. Thank you.

Reviewer #2: The article shows results of the adaptation process of the Braithwaite’s Motivational Postures (Towards Taxes) Questionnaire (MPQ) to Polish language and cultural conditions. The adaptation attempt is properly placed in the context of the Braithwaite’s (2002) concept of five tax related motivational postures. The original questionnaire is presented sufficiently and so are the steps of the adaptation. The Author conducted three studies that show construction and validation of the Polish version of MPQ. The way the Author analyses the data and present the analyses is appropriate and adequate. The only criticism relates to the use of Cronbach’s alpha as a reliability measure of subscales which consists less than 10 items (even as little as two). In such a case item-rest of the scale correlations should be included. Generally the results show clearly that the Polish adaptation of the MPQ has good psychometric properties and can be used in research on Polish samples. The use of the adaptation can extend the knowledge about tax-related attitudes and behaviors by including data from a different than western cultural and economic reality.

The manuscript is well organized and written clearly enough to be accessible to non-specialists.

Original data are available; they are deposited in appropriate repositories.

6. PLOS authors have the option to publish the peer review history of their article (what does this mean?). If published, this will include your full peer review and any attached files.

Reviewer #1: No

Reviewer #2: No

---

## [Author Response · Author response to Decision Letter 0]

21 May 2021

Dear Frantisek Sudzina,

Academic Editor

Thank you very much for the letter and insightful reviews. I have read them carefully and made corrections in line with those suggestions. I hope this work improved the manuscript.

Below, please find the detailed information about changes made in the manuscript.

1. With reference to the suggestions formulated by Reviewer #1:

- The issue of person and contextual characteristics that shape motivational postures was underlined in the Discussion section.

- Description of correlations between Individual and Social Tax Standards Scale and MPQ was corrected in order to clarify the text (line 509-510).

- In line with the suggestion the assumption regarding relations between Pleasant Game scale and belief in life as a zero-sum game was corrected (line 515). I agree that gameplaying in “Defiance in Taxation and Governance” clearly shows game playing as a Zero-sum game life view.

2. With reference to the suggestions formulated by Reviewer #2:

- I agree that in case of scales with small (less than 10) number of questions item-rest of the scale correlations are often included. However, in this paper in order to ensure the comparability with the original scale I decided to present the same indicator as in the original work. It should be noted that both versions of the scale (Australian and Polish) consist of similar (below 10) number of items. Therefore, I decided not to change the reliability measure.

3. Manuscript meets PLOS ONE's style requirements, including those for file naming.

4. Abstract – citations were removed from the abstract. Therefore, the reference list order was changed. All papers remained on the reference list.

5. Additional details regarding participants consent were provided in Participants sections in all 3 studies.

6. All typos were corrected.

I am looking forward to hearing from you.

Sincerely,

Sabina Kołodziej

---

## [Editor Report · Decision Letter 1]

26 May 2021

Validation of the Polish version of the Motivational Postures (Towards Taxes) Questionnaire

PONE-D-21-07749R1

Dear Dr. Kołodziej,

We’re pleased to inform you that your manuscript has been judged scientifically suitable for publication and will be formally accepted for publication once it meets all outstanding technical requirements.

Kind regards,

Frantisek Sudzina

Academic Editor

PLOS ONE
---

## [Editor Report · Acceptance letter]

7 Jun 2021

PONE-D-21-07749R1 

Validation of the Polish version of the Motivational Postures (Toward Taxes) Questionnaire 

Dear Dr. Kołodziej:

I'm pleased to inform you that your manuscript has been deemed suitable for publication in PLOS ONE. Congratulations! Your manuscript is now with our production department. 

Kind regards, 

on behalf of

Dr. Frantisek Sudzina 

Academic Editor

PLOS ONE